# GTExome: Modeling commonly expressed missense mutations in the human genome

Jill Hoffman[1☉], Henry Tan[2☉], Clara Sandoval-Cooper[2], Kaelyn de Villiers[2], Scott M. Reed [ID][2]*

1 Computational Bioscience, University of Colorado Anschutz Medical Campus, Aurora, CO, United States of America, 2 Department of Chemistry, University of Colorado Denver, Denver, CO, United States of America

☉ These authors contributed equally to this work.
* scott.reed@ucdenver.edu

**Data Availability Statement:** All relevant data for this study are publicly available from the Github repository (https://github.com/henrytan2/CU-Denver-Pharmacogenomics-Website) and from the GTEx Portal database (https://www.gtexportal.org/

## Abstract

A web application, GTExome, is described that quickly identifies, classifies, and models missense mutations in commonly expressed human proteins. GTExome can be used to categorize genomic mutation data with tissue specific expression data from the Genotype-Tissue Expression (GTEx) project. Commonly expressed missense mutations in proteins from a wide range of tissue types can be selected and assessed for modeling suitability. Information about the consequences of each mutation is provided to the user including if disulfide bonds, hydrogen bonds, or salt bridges are broken, buried prolines introduced, buried charges are created or lost, charge is swapped, a buried glycine is replaced, or if the residue that would be removed is a proline in the cis configuration. Also, if the mutation site is in a binding pocket the number of pockets and their volumes are reported. The user can assess this information and then select from available experimental or computationally predicted structures of native proteins to create, visualize, and download a model of the mutated protein using Fast and Accurate Side-chain Protein Repacking (FASPR). For AlphaFold modeled proteins, confidence scores for native proteins are provided. Using this tool, we explored a set of 9,666 common missense mutations from a variety of tissues from GTEx and show that most mutations can be modeled using this tool to facilitate studies of protein-protein and protein-drug interactions. The open-source tool is freely available at https://pharmacogenomics.clas.ucdenver.edu/gtexome/.

## Introduction

Missense mutations are responsible for many genetic diseases through a variety of mechanisms [1] and explain many drug side effects [2]. There are over 28 million known missense mutations that have been discovered by genome-wide association studies to date [3] and a total of over 71 million missense mutations are possible [4] making experimental study of all of them intractable. Only 17% of native human protein structures have experimental structures available [5] and missense structures are rarer still. AlphaFold is a deep learning method that predicts 3D protein structure from the amino acid sequence based on known protein structures

home/downloads/adult-gtex/bulk_tissue_expression).

**Funding:** JH was supported by the MARC U-STAR program (NIGMS T34 T34 GM096958) Research reported in this publication was supported by the National Institute of General Medical Sciences of the National Institutes of Health under award number R15GM151726 (SMR).

**Competing interests:** The authors have declared that no competing interests exist.

with similar sequences [6]. By using artificial intelligence to model entire proteomes, including that of humans [7], AlphaFold has opened new methods to study protein structure and many groups continue to improve upon the publicly available code for AlphaFold [8]. AlphaFold structures have been used to study protein-protein interactions [9], protein function [10], and small molecule docking [11, 12].

In one analysis, AlphaFold was used to model missense mutations in three proteins where a specific mutation was known to disrupt the main chain packing of the protein (including ubiquitin-associated domains of a human Rad23 protein, BRCA1, and the MyUb domain of actin motor protein Myosin VI) [13]. In those cases, AlphaFold did not capture the changes seen in experimental structures caused by the single nucleotide variation (SNV). A more systematic exploration of missense modelling has not been conducted and would be difficult and time-prohibitive given the currently available methods for generating structures. Using the publicly available cloud-based version of the AlphaFold software, called ColabFold, it takes about 45 minutes to create a single missense variant [14].

Here we describe an open-access, high throughput, web-based tool, called GTExome, for identifying and visualizing the effect of single missense mutations on the three-dimensional structure of proteins. Native protein inputs can be selected from either experimental structures or AlphaFold predicted structures. The missense mutation data comes from aggregated large scale sequencing data from 60,706 exomes available in the public database gnomAD [15]. To help focus searches on more commonly expressed proteins and to aid matching hypotheses to tissues where these proteins are common, tissue specific gene expression data from the Genotype-Tissue Expression (GTEx) database was used to scaffold access to the SNV data. This tissue-specific data is coupled in, allowing for searches based on protein expression levels measured from RNA-Seq experiments in different tissues [16]. GTExome provides lists of mutations for download that have been discovered for these commonly expressed proteins. Using GTExome, we created three-dimensional structures for 9,666 of the most common mutations in common proteins and analyzed their suitability as models.

## Results

To select genes based on expression in different tissue types, the *gtex* tab is selected (Fig 1, step 1a). The genes can be selected based on the absolute expression in transcripts per million (TPM) or as a ratio of expression in one or more selected tissues to the remaining tissues. To enter a gene directly without using tissue-specific expression data, the *exac* tab can be selected. Any gnomAD gene name entered there will lead directly to a list of SNVs for that gene (Fig 1, step 1b). Alternatively, if both a gene (by ENSG number) and specific SNV (by HGVS Consequence) are known already, one can select the *refold* tab, enter this info, and proceed directly to the mutation analysis page (Fig 1, step 1c).

In steps 4 and 5, after a specific mutation is selected by one of the three entry points, the possibility that the mutation results in a protein with a similar fold to the native is evaluated. In some cases, we expect the mutations will minimally disrupt backbone structure. In these cases, the user can create a new model for the missense variant using the same three-dimensional backbone coordinates from the experimental or modeled native protein. The user can select from the highest resolution available experimental non-SNV containing structure or from an AlphaFold model of the native protein. To accommodate the mutation, side chains in the region can be repacked within a user-defined region to provide a structure as close to the native as possible while avoiding side chain conflicts. This is accomplished using Fast and Accurate Side-chain Protein Repacking (FASPR) [17]. FASPR is used to sample the side-chain rotamers for each amino acid within the assigned radius of the missense location. Atomic

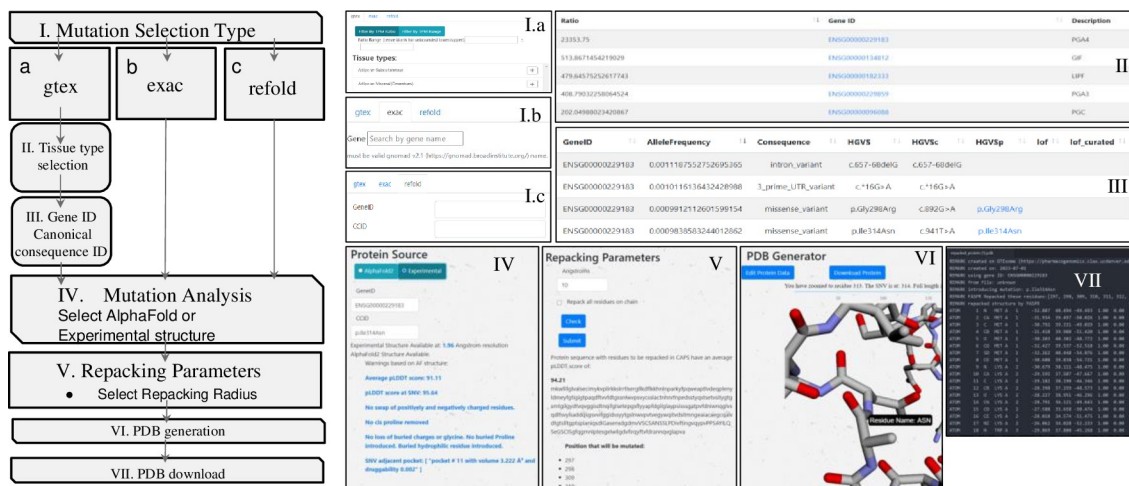

**Fig 1. Walkthrough of the GTExome website at https://pharmacogenomics.clas.ucdenver.edu/gtexome/ showing pageviews in the sequence used to generate a missense mutation model.**

interaction energies are calculated using a scoring function where the side-chain packing search is performed using a deterministic searching algorithm combining self-energy checking, dead-end elimination, and tree decomposition. An analysis of the impact on the radius that is repacked on the number of residues that end up with different side chain conformations in contrast to direct modeling using ColabFold (In the S1 File) reveals that a repacking radius of 30 Å is sufficient to allow the side chains in the region to adopt their lowest possible conformations. In contrast to ColabFold, GTExome allows for nearly instant structure creation for any of the millions of known or hypothetical missense mutations in the exome. FASPR repacks residue side chains while maintaining backbone structure. This ensures there is less change to the native structure to ensure a more accurate model for non-disruptive mutations. Users are provided with information that can help in deciding if the accuracy of the model is sufficient for their use and can select a region of the protein in which the side chain atoms are repacked. Like Missense3D [18], warnings include if there is a swapped charge in a residue, a cis proline is replaced, there is a gain or loss of a buried charge, loss of a buried glycine, a buried proline is introduced, or loss of a residue that hydrogen-bonds with another residue. Furthermore, for the 9,660 most common mutations, if the mutation is adjacent to a binding pocket as determined by fpocket [19], the volume of any pockets that contain the residue at the site of the SNV are provided. Higher pLDDT scores correlate with better correspondence in small molecule binding sites compared to experimentally determined structures [11] and more accurate binding pocket predictions [20].

If a hydrogen bond is removed by the mutation (identified if hydrogen donor atoms are within a 3.0 Å distance of hydrogen acceptor atoms) the mutation is flagged. Similarly, disruption of a salt bridge, where oppositely charged atoms are within a 3.2 Å distance is flagged. These distance values are the default parameters used in VMD to identify these types of bonds [21]. Disulfide bond presence was directly searched for using labeled ssbonds in the Protein Data Bank (PDB) file. Charge switches were identified if negatively charged amino acids (Glu, Asp) were mutated to a positively charged amino acids (His, Lys, Arg) or vice versa.

Gain or loss of buried charges were determined if either a positively or negatively charged amino acid were either replaced by or replaced a non-charged amino acid. The gain of buried proline or loss of glycine was flagged if the amino acid a residue was being mutated was a proline or glycine. The gain of a buried hydrophilic residues was identified if the mutation residue

was hydrophilic (Ser, Thr, Cys, Tyr, Asn, Gln) while the native residue is not already a hydrophilic residue and a hydrogen bond acceptor was within binding distance.

Prolines in cis configurations, if mutated will remain cis when repacked using FASPR. Torsion angles of proline were calculated to determine if they were in cis or trans configuration before mutation. A torsion angle that deviated from the range of 150° to 190° is flagged as cis.

Fpocket is used to determine residues that are a part of binding pockets [19]. Pockets are determined based on the alpha spheres present in the protein [22]. If the mutation site involved in any of the identified pockets, it is flagged. In addition, the druggability score for each pocket is provided to the user [23].

To model the mutation before repacking, the user can choose the radius around the mutation site where side chains will be repacked or select all residues to be repacked. The sequence of the protein is returned with the list of residues to be repacked listed in all caps and the mean pLDDT score of those residues provided. The repacked protein is shown to the viewer using a plotly-dash tool that includes a slider allowing zoom in on specific residues with the missense mutation highlighted (Fig 1, step 6). Finally, the user can select to download a structure file or to go back to re-edit the input parameters (Fig 1, step 7).

In a sample trial that highlights one use of GTExome, we examined the genes that had an expression ratio at or above 1 relative to expression in all other tissues combined for each tissue in GTEx (Fig 2). Of the 9,666 genes identified, 78% of the SNVs were rare, with an allele frequency of <0.1. The mean pLDDT score across all the mutations was 70.43 at the site of mutation. No mutations resulted in a loss of disulfide or buried glycine. Across all mutations studied, a change in charge from positive to negative or negative to positive occurred in 3% of the SNVs, loss of a cis proline in 7%, loss of a salt bridge in <1%, and loss of a possible buried

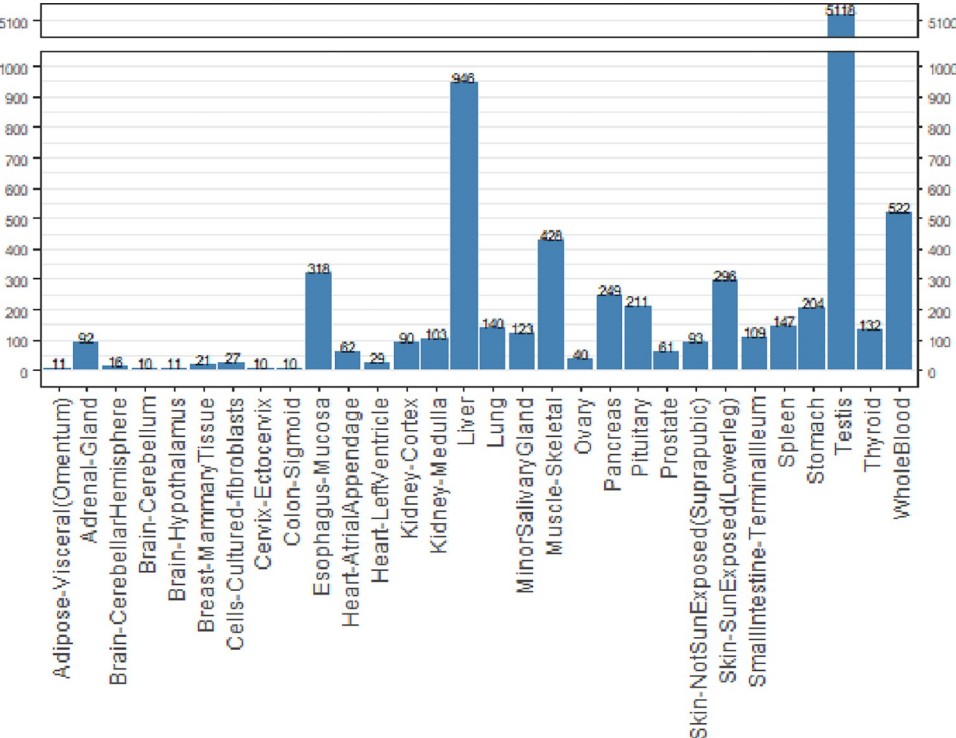

**Fig 2. Number of missense SNVs of the 9,660 examined found by tissue type from GTEx for the 9,660 SNVs examined across the GTEx tissues with 10 or more SNVs above ratio threshold.**

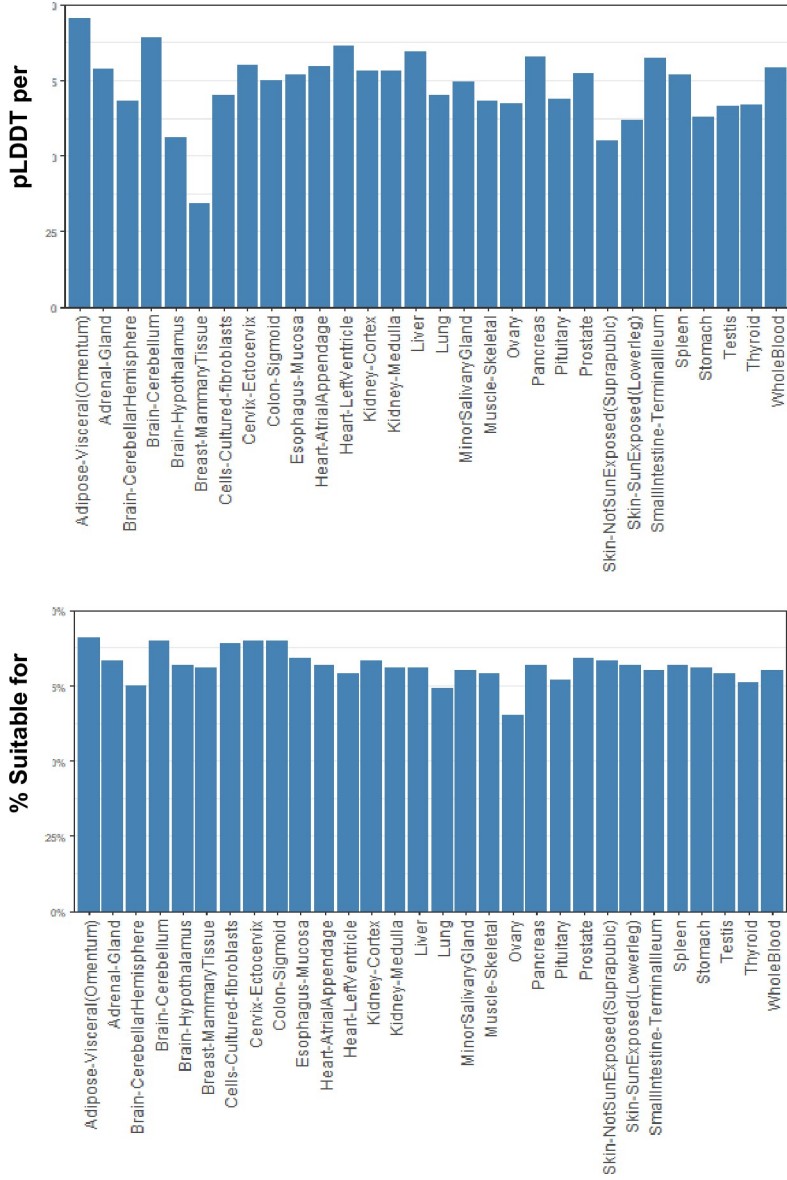

**Fig 3.** Top) Percentage of SNVs of the 9,660 examined found suitable for modeling, and bottom) mean pLDDT score at site of SNV by tissue type (for tissues with 10 or more SNVs above ratio threshold).

hydrogen bond in 7%. Overall, we classified 80% of the SNVs as AlphaFold suitable based on not having any of these deleterious changes that are unlikely to be modeled correctly by applying FASPR to the native structure. Limiting to proteins with a pLDDT score at the SNV site above a stringent threshold of 90 would limit this further. The average pLDDT per tissue type for tissues with 10 or more proteins and the percentage suitable by tissue type are shown in Fig 3 and the rates of cis proline loss, salt bridge loss, or changes to buried hydrogen bonds or charge swaps are show in (the S1 File and S1–S5 Figs).

AlphaMissense is a newly available tool that predicts the pathogenicity of 71,140,163 feasible missense mutations that could occur in the human exome. We compared our trial set of mutations to predictions made by AlphaMissense and found that 88% of the 9,660 are classified as benign and 6% as pathogenic. This is higher than the 57% reported as benign in

AlphaMissense for all potential missense mutations. This supports the idea that SNVs that are expressed in tissues are under more selective pressure and that this filtering method favors mutations that are likely to produce non-pathogenic proteins. In contrast we found that the median COSMIS value for the 3493 mutations of this study set that were in the COSMIS database was 0.3882 for values derived from AlphaFold structures and 0.4819 for PDB derived structures. Both values are higher than the -0.47 reported for median SNV across the entire proteome [24]. This higher value indicates that the three-dimensional region surrounding these mutations are under less evolutionary constraint than average.

We also compared the 9,660 selected mutations against a list of mutations shown to cause adverse drug reactions that have been shown to be clinically relevant [2]. We identified 14 missense SNVs that overlap between our selected mutants and those in the study. These SNVs span 6 different genes of the 12 studied by Swen et al., all of which are available as AlphaFold models in GTExome. The SNV sites have an average pLDDT score of 95 and 10 out of 14 are rated as suitable for modeling based on the type of mutation (In the S1 File and S1 Table).

Currently, modeling the effects of missense mutations without experimental structures can be done by running AlphaFold directly on the sequence or running it remotely using Colab-Fold. We compared 47 SNVs modeled using GTExome and ColabFold and found very similar structures. The C-$\alpha$ RMSD values between the two structures after performing an alignment in Chimera had a median value of 0.52 Å for a range of protein sizes and types of mutations (In the S1 File and S2 Table and S1 Fig).

An Application Programming Interface (API) endpoint is provided on the website to developers through a registration process to allow direct calls to many of the backend APIs that run the website. One access point provides the PDB file name, chain ID, and best resolution of an experimental structure for a given the ENSG number. A second API takes a geneID and the HGVS consequence (CCID number from the ExAC browser) as input and returns the pLDDT score at the SNV, the mean pLDDT score for all residues in the protein, the number, volume, and druggability of any pockets adjacent to the SNV, the name of the AlphaFold file, a recommendation on whether the AlphaFold structure is suitable for modeling, and the individual results of each amino acid check (as provided in the web interface). The third API takes an input geneID, CCID, and repacking radius and returns the list of residues, the sequence length, the sequence of the protein with residues within the given radius in all caps, and the mean pLDDT score for the residues that would be repacked at that radius.

To compare predicted structures to actual experimental structures containing the same mutation, we searched the Structure Integration with Function, Taxonomy and Sequence (SIFTS) database [5] for files in the PDB that match the Uniprot number for the given parent protein for each of the 9,660 mutations. Then we examined each chain in each model for these PDB files to see if they had the same mutation. We found a total of 8 SNVs in our set that had verified matching experimental structures determined by x-ray crystallography in the PDB with high overlap to the modelled structure and that contain the same mutation. We examined the C-$\alpha$ RMSD values in comparing the two (In the S1 File and S3 Table) after using matchmaker in Chimera to align the two structures. The structures were similar in each case with a mean RMSD of 0.55 Å for the overall structures, the region where the SNV occurred was very similar, and the side chains at the SNV had small rotational differences in only a few cases (Fig 4).

## Materials and methods

The GTExome web application was written using the Python-based Django framework. The backend is comprised of a mySQL relational database and the frontend rendered with a

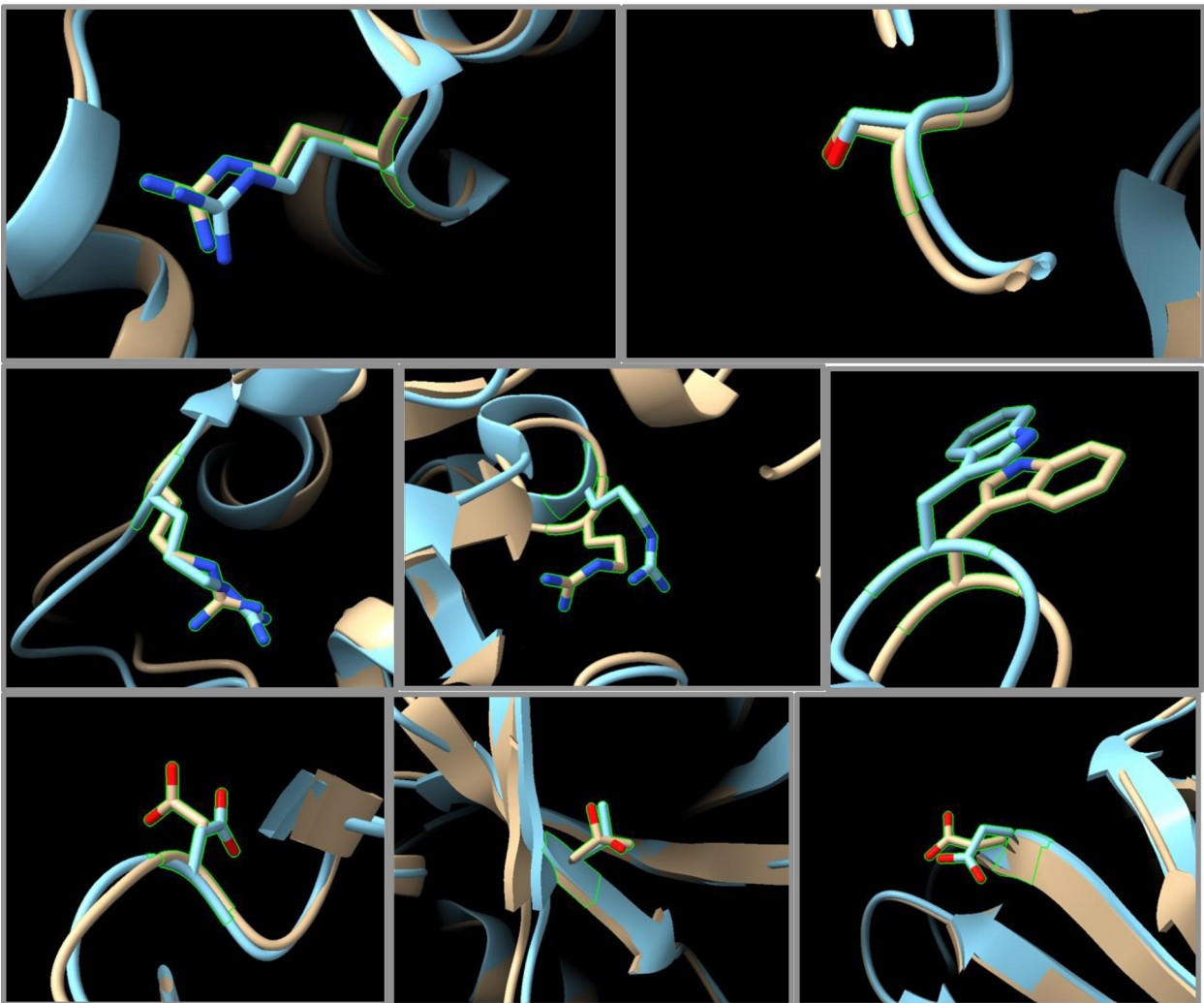

**Fig 4. Comparison of experimental x-ray structures (blue) that contain a missense mutation that also appears in our survey.** Structures created on GTExome using AlphaFold input modified with FASPR using at a radius of 30 Å (brown). Protein ascension numbers and SNV: a) ENST00000263126:Gln250Arg, b) ENST00000287777:Asn345Ser, c) ENST00000324071:Lys262Arg, d) ENST00000287777:Cys617Arg, e) ENST00000260302:Ser455Trp, f) ENST00000341010:Ala51Asp, g) ENST00000341010:Ile44Thr, h) ENST00000358834: Glu414Asp.

combination of Django, Javascript, JQuery, and Vue for handling reactive elements built alongside a second web application, Metabolovigilance [25].

Three ways to input protein mutation selection are provided; *gtex*, *exac*, and *refold*. For the *gtex* tab, data from GTEx is stored locally with expression level for all proteins. The counts are available in TPM for each gene and stored in the database for the 54 tissue sites and 2 cell types obtained from nearly 1000 individuals in the GTEx project. Gene lists are created by either a user input range of TPM counts for genes in a tissue or combination of tissues or from a ratio where the numerator is the sum of expression in selected tissues and the denominator is the sum of the remaining tissues. The *exac* tab can be used to get a list of all SNVs for a single protein if the gene name is known. For *gtex* and *exac* tabs, a button is provided to filter the results to display only missense mutations. The *refold* tab can be used if the specific gene ID and mutation ID is known. A search box on each page provides additional searching and filtering methods.

Contact Set Missense Tolerance (COSMIS) code was modified to be used to determine the experimental structure with the best match to the sequence provided. COSMIS was originally used to quantify constraint of mutations on proteins [22]. The first part of their process was repurposed, which utilizes the SIFTs database [5]. SIFTs was used to search for an experimental structure based on the geneID with the highest sequence overlap and the highest resolution of the structures only if the SNV site was located within the structure file [5]. AlphaFold models created using AlphaFold2 are available on the GTExome server for 98.5% of human proteins. Proteins longer than 2,700 amino acids are excluded [7]. If both AlphaFold and experimental structures are available, a user can select which structure to use. The pLDDT score for AlphaFold structures and the resolution of experimental structures are provided to the user. pLDDT scores are a prediction of global distance test (GDT) based on backbone RMSD values using the following equations.

$$GDT_{TS} = \frac{GDT_{P_1} + GDT_{P_2} + GDT_{P_4} + GDT_{P_8}}{4}$$

Where $GDT_{Px}$ are the percentage of C-α atoms with cutoffs of 1, 2, 4, and 8 Å RMSD where the RMSD between predicted and actual values of distance in the x, y, and z coordinates is calculated as:

$$RMSD_{a,b} = \sqrt{\frac{1}{n}\sum_{i=1}^{n}(a_{ix} - b_{ix})^2 + \left(a_{iy} - b_{iy}\right)^2 + (a_{iz} - b_{iz})^2}$$

Both the average pLDDT score (the average pLDDT score of each residue in the protein), and pLDDT score at the site of the mutation are presented. For proposed repacking, the mean of the pLDDT score of those residues that would be repacked is provided.

## Conclusion

Missense mutations in the human exome play a critical role in understanding human health, however experimental structures of them are rare. We found 8 structures while examining close to 10,000 proteins that are commonly expressed in healthy adults. This underscores the need for accurate models of these structures. In this publication, a fast, high-throughput tool to model protein missense mutations is described. This tool increases the availability of mutated model proteins as few experimental structures have been determined experimentally [5]. This tool makes 98.5% of human proteome available through either AlphaFold and/or experimental structures. Users can choose their preferred parameters for the model and are presented with information about the suitability of the model being created. The output structures can be used as a starting point for more advanced structure modeling for docking studies. This fast tool provides the best available structures to users and can be updated as better models become available.

Describing if the location of the missense mutation is a part of a binding pocket allows facilitating selection of models for studying the effects of missense mutations on drug-protein interactions. With this information, predicted structures can be used later in docking studies to understand possible side effects associated with missense mutations, or to research protein–protein interactions. A recent drug docking study using AlphaFold models [12] reported a median RMSD of 2.9 Å between AlphaFold models and corresponding experimentally determined structures, significantly more accurate than the RMSD of 4.3 Å for traditional models indicating that despite imperfections in the side chain structures, the cavities are generally well represented. For virtual screening, this may be sufficient to identify possible hits.

In contrast to the three proteins reported previously [13] which may have been selected precisely because they were anticipated to be difficult to model with AlphaFold, here we selected proteins based on expression levels and found high similarity between experimental and predicted structures and high similarity to models created directly by ColabFold. The comparison to experimental structures provides evidence of similarity between modeled and experimental structures with the important caveat that the structures examined were deposited in the PDB between 1993 and 2015 which pre-dates AlphaFold and were therefore likely part of the training set used to build the predictor.

## Supporting information

**S1 File.**
(DOCX)

## Author Contributions

**Conceptualization:** Scott M. Reed.

**Data curation:** Jill Hoffman, Henry Tan, Clara Sandoval-Cooper, Kaelyn de Villiers, Scott M. Reed.

**Software:** Jill Hoffman, Henry Tan, Scott M. Reed.

**Visualization:** Jill Hoffman.

**Writing – original draft:** Jill Hoffman, Scott M. Reed.

**Writing – review & editing:** Scott M. Reed.

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
