## [Decision Letter · Decision Letter 0]

24 Mar 2024

PONE-D-24-03381GTExome: Modeling commonly expressed missense mutations in the human genomePLOS ONE

Dear Dr. Reed,

Thank you for submitting your manuscript to PLOS ONE. After careful consideration, we feel that it has merit but does not fully meet PLOS ONE’s publication criteria as it currently stands. Therefore, we invite you to submit a revised version of the manuscript that addresses the points raised during the review process.

We look forward to receiving your revised manuscript.

Kind regards,

Yong Wang

Academic Editor

PLOS ONE

Journal Requirements:

"JH was supported by the MARC U-STAR program (NIGMS T34 T34 GM096958)

Research reported in this publication was supported by the National Institute of General Medical Sciences of the National Institutes of Health under award number R15GM151726 (SMR)."

4. Please expand the acronym “NIH” (as indicated in your financial disclosure) so that it states the name of your funders in full.

5. Please ensure that you refer to Figure 3 in your text as, if accepted, production will need this reference to link the reader to the figure.

Reviewers' comments:

Reviewer's Responses to Questions

**Comments to the Author**

1. Is the manuscript technically sound, and do the data support the conclusions?

Reviewer #1: Yes

2. Has the statistical analysis been performed appropriately and rigorously? 

Reviewer #1: N/A

3. Have the authors made all data underlying the findings in their manuscript fully available?

Reviewer #1: No

4. Is the manuscript presented in an intelligible fashion and written in standard English?

Reviewer #1: Yes

5. Review Comments to the Author

Reviewer #1: In this paper, the authors have developed a web-tool that provides homology or alphafold models on which various features of human missense mutants are presented for analyses. The purpose of this tool sounds fine and the product would be useful for other researchers, if it is continuously maintained and updated. To this reviewer’s understanding, this manuscript has been revised already based on the comments from two reviewers, and the comments were proper and acceptable. Thus, this reviewer is basically positive for publishing this paper. Only a few minor comments, which should be considered in preparing a final version of the manuscript, would be raised.

1) P8: “feasible missense mutations are possible” This reviewer could not understand what “feasible” mean. It should be explained in the text.

2) P11: “the COSMIS p-value for the 8,780 mutations of this study set that were in the COSMIS database was -0.0156 for p-values derived from AlphaFold structures and 0.02 for PDB derived structures.”　Why can a p-value be negative? This sentence is difficult to follow. COSMIS is first spell-out　(Contact Set Missense Tolerance) in p15.

3) P12: “determined by x-ray crystallography“ and P15: “Figure 4. Comparison of experimental x-ray structures (blue)…” The PDB codes should be provided.

4) P13: “Figure 2. Number of missense SNVs of the 9,660 examined found by tissue type from GTEx for the 9,660 SNVs examined across the GTEx tissues with 10 or more SNVs above ratio threshold.” Which part of this is the figure title and which part is the legend?

5) P16: “using software such as Dynamut.” What is Dynamut? It should be briefly explained.

6) P8:” AlphaFold was used to model missense mutations in three proteins where a specific mutation was known to disrupt the main chain packing of the protein” and P17: “In contrast to the three protein missense mutations reported previously…“ This reviewer assumed these sentences are referring to same entities. If so, it is very confusing because the former is protein and latter is mutation, and also because these sentences are separated in the context. The genes(proteins) and mutations should be explicitly mentioned.

6. PLOS authors have the option to publish the peer review history of their article (what does this mean?). If published, this will include your full peer review and any attached files.

Reviewer #1: **Yes: **Tsuyoshi Shirai

---

## [Author Response · Author response to Decision Letter 0]

25 Apr 2024

Subject: Response to Reviewer Comments - Revised Submission of Manuscript PONE-D-24-03381

Dear Dr. Yong Wang,

We are grateful for the opportunity to resubmit our manuscript, "GTExome: Modeling Commonly Expressed Missense Mutations in the Human Genome," to PLOS ONE. We appreciate the thorough review and constructive comments provided by the academic editor and the reviewer. We have carefully addressed all the points raised and believe these changes have significantly improved our manuscript.

As requested, we have uploaded three separate files: a rebuttal letter (this document), a marked-up copy of our manuscript highlighting the changes made (Revised Manuscript with Track Changes), and an unmarked version of the revised paper (Manuscript).

Below, we summarize how we addressed the major points in your request and the reviewer's comments:

1. PLOS ONE's Style and File Naming Requirements: We have revised our manuscript to meet the style requirements as outlined and have ensured all files are correctly named according to the provided templates.

2. Code Sharing Guidelines Compliance: In line with PLOS ONE's code sharing policy, we have made all author-generated code available without restrictions.

3. Financial Disclosure and Funder's Role: We have amended our financial disclosure to fully spell out "National Institutes of Health". 

4. Reference to Figure 3 and Supporting Information Captions: We have ensured that Figure 3 is referred to appropriately within the text and have ensured all figures have captions in the Supporting Information files as instructed.

5. Reference List Accuracy: The reference list has been thoroughly reviewed and updated to ensure completeness and correctness.

Addressing Reviewer's Comments:

1. The word "feasible" modifying missense mutations has been removed as unnecessary and confusing. 

2. We have corrected our COSMIS analysis and changed the text to reflect our updated evaluation to say, “… median COSMIS value for the 3493 mutations of this study set that were in the COSMIS database was 0.3882 for values derived from AlphaFold structures and 0.4819 for PDB derived structures.” This revised statement clarifies we are reporting COSMIS values not COSMIS p-values and that we re reporting median values. The conclusion remains unchanged.

3. PDB codes for structures determined by x-ray crystallography have been provided as requested in table S3.

4. A correction has been made to the location of the citation of Figure 2 within the paragraph.

5. The sentence mentioning Dynamut software has been trimmed and the reference to this software removed as it did not add to the clarity of the manuscript.

6. The confusion between proteins and mutations discussed on pages 8 and 17 has been clarified, with specific proteins explicitly mentioned.

We trust that these revisions address the concerns raised during the review process. We thank you again for the opportunity to improve our manuscript and for considering it for publication in PLOS ONE.

Kind regards,

Prof. Scott Reed

Department of Chemistry

University of Colorado Denver

1151 Arapahoe St.

Science Building 4131

Denver, CO 80217-3364

scott.reed@ucdenver.edu

(303) 315-7634

---

## [Editor Report · Decision Letter 1]

29 Apr 2024

GTExome: Modeling commonly expressed missense mutations in the human genome

PONE-D-24-03381R1

Dear Dr. Reed,

We’re pleased to inform you that your manuscript has been judged scientifically suitable for publication and will be formally accepted for publication once it meets all outstanding technical requirements.

Kind regards,

Yong Wang

Academic Editor

PLOS ONE
---

## [Editor Report · Acceptance letter]

9 May 2024

PONE-D-24-03381R1 

PLOS ONE

Dear Dr. Reed, 

I'm pleased to inform you that your manuscript has been deemed suitable for publication in PLOS ONE. Congratulations! Your manuscript is now being handed over to our production team.

Kind regards, 

on behalf of

Dr. Yong Wang 

Academic Editor

PLOS ONE